# Peculiarities of Fe and Ni Diffusion in Polycrystalline Cu

**DOI:** 10.3390/ma16030922

**Published:** 2023-01-18

**Authors:** Alexey Rodin, Ainur Khairullin

**Affiliations:** Department of Physical Chemistry, National University of Science and Technology “MISiS”, NUST “MISiS”, 4, Leninsky pr-t, Moscow 119049, Russia

**Keywords:** diffusion, grain boundary, copper, segregation

## Abstract

The peculiarities of Ni and Fe diffusion in polycrystalline Cu and Cu-Fe alloys at a 650–750 °C temperature range were studied. It was shown that the bulk diffusion coefficients were in a good agreement with the available literature data obtained for different temperature concentration ranges. The obtained values of the Ni GB diffusion triple product could be described as $s\delta D_{b}=8.8\times 10^{-13}\times exp\left(\frac{-165\;kJmol^{-1}}{RT}\right)m^{3}s^{-1}$. The obtained values were an order of magnitude less than the values obtained by the radiotracer method. This fact was explained by the negative segregation of Ni at the Cu GB. A stronger negative segregation of Fe in the copper GB reduced the penetration depth of iron along the GB because of an additional (negative) “driving” force; this explained why there was no advanced grain boundary diffusion. Another peculiarity of the Fe diffusion that was confirmed was the significant supersaturation of the Cu-based solid solution near the Fe/Cu interface.

## 1. Introduction

Diffusion in grain boundaries (GBs) has been of great interest for the last 70 years due to a significant (several orders of magnitude) difference between the GB and bulk diffusion coefficients. This effect is especially pronounced at moderate and low temperatures (below 0.7 T_m_, where T_m_ is the melting point); the mass transport in polycrystals associated with the flux of the diffusant in the GB becomes dominant over the direct flow through the grain volume. Over these years of research, various methods to study grain boundary diffusion and approaches to their accurate mathematical treatment have been developed, so that we can now postulate that a foundation for predicting the behavior of various elements in polycrystalline systems has already been constructed.

The attempt to enlarge the approaches—developed for bulk diffusion to GB diffusion, but taking into account the peculiarities of this object—looks natural. For example, the basic method for the experimental determination of diffusion parameters is the radiotracer method (see e.g., [1]), which can be described as a diffusion of the tracer in its dilute solution in a matrix. A matrix can be a one-component system or a multicomponent system and the concentration dependence of the diffusion coefficient can be obtained by measuring parameters at different concentrations of the alloying elements. A more complex approach, both in creating a physical model and in the mathematical description, is based on the study of the mutual diffusion of elements in a wide concentration range because the diffusion of two or more components (generally speaking, not independent) is accompanied by a possible phase formation, pore formation, shift of interface and other effects that must be taken into account. The situation with grain boundaries is even more complicated because they are not the same, even within one sample. Their properties are very sensitive to the angle of grain misorientation, the presence of impurities, the possibility of boundary migration and other effects [2]. In this sense, a comparison of the grain boundary diffusion data obtained by different authors requires a deep analysis, especially if different methods for the determination of parameters are used. It is necessary to underline that in the practically important temperature regime of moderate temperatures (close to 0.5 T_m_), the grain boundary diffusion can be described by a so-called triple product *P* = *sδD_b_*, where $s={\left(\frac{c_{b}}{c}\right)}_{x=\pm \delta /2}$ is the GB enrichment factor (ratio of the GB and bulk concentrations), *δ* is the grain boundary width and *D_b_* is the GB diffusion coefficient. This is because the bulk diffusion determines not only the direct flux from the surface into the grain, but also the flux from the GB to the grain [1,2]. It is evident that the greater the GB segregation factor, the greater the triple product and the deeper the penetration of the diffusant along the GB. 

In this paper, we present a comparative analysis of the results of the study of the bulk and GB diffusion obtained by different methods (the methods of radiotracer isotopes and a microprobe analysis (EPMA)) for iron and nickel in copper. As additional data to the discussion, the results of the computer modeling for these systems as well as for a cobalt–copper system were used.

The choice of the system was based on scientific interest of a system with negative GB segregation (segregation factor s < 1) [3,4], but Cu-Fe alloys are of a practical interest as a soft magnetic material [5] as well as being conductive materials with enhanced mechanical and wear-resistant properties [6]. Fe-based coatings allow the improvement of wear resistance for several tools [7,8]. The stability of the microstructure and properties at elevated temperatures are connected to the diffusion properties of polycrystalline alloys. 

The bulk and GB diffusion of the chosen systems have been studied in a number of works. One can observe a good agreement between the bulk diffusion data both for Ni [9,10,11,12,13,14] and Fe [9,15,16,17,18] diffusion in Cu. Regarding the GB diffusion, the difference in the results is significant. 

Triple product values for Ni in Cu obtained by autoradiography [19,20,21] demonstrate a difference of two orders of magnitude. The most reliable data, obtained by the radiotracer technique [22], provided intermediate values; however, the data obtained by an electron probe microanalysis (EPMA) [23] obtained an order of magnitude with higher values than [22]. This is surprising, because radiotracer methods typically provide the characteristics for the fastest GB [24].

According to [25], the triple product of Fe diffusion is P=9.1×10−12exp{−121kJ/molRT}, but this could only be determined in a temperature range of 949–1247 K, which is more than 0.7 T_m_. Our results [26,27,28] demonstrated the absence of a concentration excess near the GB in comparison with the bulk at high (above 1073 K) and low (below 873 K) temperatures; thus, the GB diffusion parameters could not be determined. On the other hand, the triple product recalculated from the growth rate of cobalt–iron particles at the GBs of a copper alloy at 823 and 873 K showed the absence of any anomalies [29], providing a reasonable value for the GB triple product close to Cu self-diffusion.

In order to ascertain the possible reasons of these contradictions, a diffusion study with an enlarged time–temperature range was conducted for an Fe-Cu system. Special attention was paid to the correct treatment of the results; thus, the experiment was conducted both for the cross-section and the foils at the same temperatures. In order to study the possible effect of Fe on the diffusion characteristics of the grain boundaries in Cu, the effect of the alloying of Cu by Fe was studied by using Fe and Ni as diffusants.

## 2. Materials and Methods

An energy dispersive X-ray microanalysis (EDX) was used to determine the concentration of the diffusing elements near the grain boundary and far from it. It allowed us to obtain the data by a direct comparison of the penetration depths of the elements. If there was a significant difference in the penetration depths, there was the possibility of separately determining the parameters of the GB and bulk diffusion. The accuracy of this method does not exceed 0.1%; therefore, only data above 0.2–0.3% were taken into account for the analysis. 

For the research, copper of 99.995% purity, iron of 99.99% purity, chemically pure (better than 99%) nickel and iron sulfates (NiSO_4_·7H_2_O and FeSO_4_·7H_2_O) were used.

The Cu-Fe alloys were prepared by dissolving the master alloys (containing 1.25% iron) in liquid copper at a temperature of 1200 °C in Ar with the small addition of a H_2_ atmosphere. The melts were kept for 4 h at this temperature and cooled down by removing the quartz reactor from the furnace under the conditions of a weak Ar flux.

The ingots (diameter of about 20 mm) were kept at a temperature of 1050 °C in an Ar atmosphere for 20 h. They were cut into discs of approximately 3 mm thickness; these were carefully cleaned, thinned by deformation (70%) and polished. After that, recrystallization annealing was carried out in a hydrogen atmosphere at 1000 °C for 1 h.

The iron content in the samples was determined by atomic emission spectroscopy with inductively coupled plasma. Two types of samples with 0.4 at% and 0.3 at% were obtained.

An electrochemical deposition from sulfate aqueous electrolytes was used to produce the sample for the diffusion study. The electrolyte for the nickel deposition comprised nickel sulfate (250 g/L), boric acid (30 g/L) and sodium chloride (10 g/L) as well as distilled water. The Fe deposition comprised iron sulfate (250 g/L), potassium sulfate (150 g/L) and oxalic acid (4 g/L). The time was selected in order to obtain a 20 μm-thick layer. The samples were annealed in quartz ampoules and evacuated to a 10^−3^ mm Hg vacuum. 

Another type of sample was an 18 μm-thick Cu foil with Fe on one side, as produced in [23,24]. Typical views of the diffusion couple cross-section (points 1–5 correspond with the bulk diffusion measurements and points 6–14 are the GB diffusion measurements) and the reverse side of the annealed foil are presented in Figure 1.

## 3. Results

### 3.1. Ni Diffusion

The diffusion of nickel into pure copper was studied at temperatures of 650 and 750 °C; with the alloys containing 0.3 and 0.4% iron, the temperatures were 650, 700 and 750 °C. Typical concentration profiles at the GB and in the bulk are presented in Figure 2.

The data on bulk diffusion were described by the erf-like equation (solution of the diffusion equation with the initial condition corresponding with a layer of finite thickness *a* and the diffusion coefficient *D* being constant):(1)$$C\left(x,\;t\right)=\frac{C_{0}}{2}\;(\mathrm{erf}\left(\;\frac{x+a}{2\sqrt{Dt}}\;\right)+\mathrm{erf}\left(\;\frac{x-a}{2\sqrt{Dt}}\;\right))$$

For the data on the grain boundary diffusion, a modified Whipple solution [30] was used: (2)$$C_{b}=C\left(x,t\right)+C_{b1}\left(x,y,t\right)\;C_{b1}\left(x,y,t\right)=\frac{C_{0}*\left(x-a\right)}{4\sqrt{\pi Dt}}\overset{\Delta }{\underset{1}{\int }}\frac{1}{\sigma^{3/2}}*\exp\left[\frac{{\left(x-a\right)}^{2}}{4\sigma Dt}\right]*\mathrm{erfc}\left[\frac{1}{2}{\left(\frac{D_{b}-D}{D_{b}-\sigma D}\right)}^{0.5}*\left[\frac{y-\frac{\delta }{2}}{\sqrt{Dt}}+\frac{2\left(\sigma -1\right)D\sqrt{Dt}}{s\delta D_{b}}\right]\right]d\sigma \;$$

This was essential because the Whipple solution is valid for the case of the boundary condition corresponding with a constant concentration on the surface. In our case, this condition was not fulfilled.

The obtained values of the bulk and grain boundary diffusion (averaged from at least 5 different profiles) parameters are presented in Table 1. 

The mean square displacement could be estimated as 10–20% for each *D* obtained from the given profile and around 30% for the averaged value at a given temperature. Thus, we observed only a small difference in the values of the GB diffusion for the pure Cu and the Cu-Fe alloys. The temperature dependences for the Ni bulk diffusion coefficient and the *P*-value are presented in Figure 3 in comparison with the data from [9,10,11,12,13,14,19,20,21,22]. It could be seen that the values of *D_Ni_* slightly differed from the other data; *P_Ni_* was significantly smaller than the data obtained in [22,23]. Notably, the Wipple approach [30] was used directly whereas in [23], the data were recalculated from the values obtained with the use of the Fisher solution for a quasi-stationary approximation [31]. 

The temperature dependence of the GB diffusion triple product (assuming the absence of a difference between the data for pure Cu and Cu-Fe alloys) could be expressed as follows:(3)$$s\delta D_{b}=8.8\times 10^{-13}\times exp\left(\frac{-165\;kJmol^{-1}}{RT}\right)m^{3}s^{-1}$$

### 3.2. Results of Bulk and Grain Boundary Diffusion of Iron into Copper and Copper-Based Alloys

The diffusion of iron into pure copper was studied at three temperatures: 650 (138 h), 700 (24, 96, 192 and 312 h) and 750 °C (30 h). The annealing time was varied at one temperature to estimate the concentration change near the iron/copper interface. For the copper–iron alloys, the annealing times were 138 and 190 h (650 °C), 49 and 72 h (700 °C) and 30 h (750 °C). The typical concentration profiles obtained from the EDX analysis are shown in Figure 4.

The concentration was expressed as the difference between the measured value and the concentration in the matrix (0, 0.3 and 0.4 at% for the different samples). Vertical lines showed the distance corresponding with the foil thickness (18 μm), taking into account the typical depth of the analysis (3 μm).

The maximum concentration (concentration at the Fe/Cu interface) was obtained by approximating the diffusion profile to a zero depth, as posited in [26,27]. To determine the *D_Fe_* and *Cs*, the erfc-like equation was used:(4)$$C\left(x,\;t\right)=C_{S}\;\mathrm{erfc}\left(\;\frac{x-a}{2\sqrt{Dt}}\;\right)$$

It is important to note that *C_S_* was almost the same for all times of annealing at a given temperature (see Table 2); thus, Equation 4 could be used. On the other hand, it decreased with an increase in the temperature (Table 3). This indicated that if one could usually use *Cs* as a solubility at this temperature ($C_{0})$, here, it was only a value corresponding with a stationary regime. 

Note the two features of the obtained profiles:The measured values of the concentration profiles were significantly above the solubility limit at the temperatures of annealing (the solubility at different temperatures is given in Table 3).The profiles near and far from the grain boundary were practically the same and a matrix concentration was reached at the same depth.

The level of iron supersaturation in the copper near the surface was determined as the ratio $C_{S}/C_{0}$. At a temperature of 650 °C, the supersaturation was 15; at 700 °C, it was 9 and at 750 °C, it was 5. The value of supersaturation decreased not only due to an increase in solubility, but also due to a decrease in *Cs*.

As it was impossible to determine the parameters of the grain boundary diffusion in the absence of a difference in the profiles of the bulk and GB diffusion, only the average values of the bulk diffusion coefficients were determined, which are given in Table 3 and Figure 5. Additionally, data obtained earlier [27] are presented. We noted that the results were in a good agreement with the previous data.

In order to compare the results with the measured values of the concentration on the opposite side of the copper foils annealed for different times, the concentrations of Fe corresponding with 18 μm were calculated (Table 4). The *Cs* values were also taken from our measurements and, as one can see, the calculated results were in a good agreement with the experimental ones for a short time of annealing. 

## 4. Discussion

According to the presented results, the used method obtained a reasonable agreement for the bulk diffusion data for both systems. For the case of Ni diffusion, the concentration profiles were quite symmetric, with the interface corresponding with its initial position. The slight non-symmetry could be explained by the concentration dependence of the diffusion coefficient. One could expect a notable difference in the diffusion coefficients at different concentration ranges because of the significant difference in the melting temperatures of copper and nickel (1084 °C and 1455 °C, correspondingly). However, as shown in [33], the concentration dependence of the diffusion coefficient was significant at a high temperature only. For the temperature range 900–1000 °C, the difference in the mutual diffusivity for different Ni concentrations was almost 100 times; at 850 °C, it was 10 times and at 730 °C, it was only 3 times and could be observed at a narrow concentration range near 80% Ni. For other concentration ranges, the diffusion coefficient was of the same value. The concentration profiles obtained in the present work confirmed this fact. The Fe diffusion coefficient was also close to the literature data. In both cases, the bulk diffusion coefficients did not depend on the preliminary alloying of the copper with iron.

Regarding the Ni GB diffusion, a significant difference could be seen, with the experimental data measured in a similar way [23]. First, we noted that it was necessary to modify the Whipple equation for the calculation because the conditions did not fully correspond with the diffusion from a source, with a constant concentration for both the bulk and GB diffusion. A significant difference from the radiotracer results [22] was reasonable because the treatment of the tails of the concentration profiles provided the diffusion coefficient for the fastest GB but not the average [24]. Taking into account that the scattering of the GB diffusion parameters can be 1–2 orders of magnitude [34], the large scatter of our results and the difference with the data, as mentioned above, appeared to be natural. We also observed a significant difference in the GB diffusion effective activation energy. The calculated value (160 kJ/mol) was significantly higher than the typical value of ½ E (with half the activation energy of the bulk diffusion being around 100 kJ/mol) and the values obtained in other works (e.g., 91 and 74 kJ/mol in [13] for Cu samples of different purity). The effective activation energy of GB diffusion consists of two terms connected with the activation energy of diffusion and the enthalpy of segregation. This is the reason why, typically, the activation energy of chemical GB diffusion is less than for self-diffusion (the enthalpy of adsorption is negative). However, the results of a number of studies on equilibrium adsorption in copper–nickel systems allowed us to conclude that copper positively segregated in the entire concentration range; thus, the segregation of Ni was negative [3]. It opposes the calculation from the diffusion results, which provided a positive segregation [22]. The difference in the results may have been due to the fact that the radiotracer method requires relatively short times. The determined enrichment coefficient of the GB did not correspond with an equilibrium state or even a quasi-stationary regime. The diffusion process can be described as the movement of the diffusing element along the GB and atoms partly enter the volume from the grain boundary. Thus, on the GB, the concentration is constantly greater than in the adjacent bulk. Studies with non-radioactive isotopes require significantly longer times and the established ratio is closer (although it may still be different) to that of the equilibrium.

The absence of the effect of pre-alloying was obtained both in this work and in [22]. It is clear that diffusion, especially isotopic diffusion, is sensitive to changes in the GB structure; thus, it is natural to conclude that the structure of the GB and its kinetic characteristics do not change under the influence of such alloying. In [25], it was concluded that iron negatively segregated and exited from the grain boundaries (or the grain boundary moved from the line of Fe atoms). The same conclusion can be reached by analyzing the results of the computer modeling performed in [35]. Despite the different models taken for the description of the interaction between atoms, the general effect is the same. The close effect for the diffusion of Co in Cu provides the same conclusion that a negative segregation significantly slows down GB diffusion; thus, it cannot be ascertained if the concentration in the bulk is relatively high or not. 

There are no direct measurements of iron (as well as cobalt) segregation on the surface and at grain boundaries. However, the isotherms of the surface and grain boundary energy of cobalt and iron in copper obtained in [4,36] directly indicated the growth of surface energy at the initial section of the curve, which, according to the Gibbs adsorption equation, indicates a negative adsorption. The enrichment coefficient estimated from this data was approximately equal to 0.1–0.3. As was shown in a modified model of GB diffusion, taking into account the surface energy gradient as an additional driving force [37], the additional term was only important in the case of negative segregation and only in the case of a relatively high concentration level. As was shown, the concentration of iron reached significantly higher values than the solubility limit. The reason for the formation of a supersaturated solid solution was the absence (or almost an absence) of Cu diffusion in iron. In contact with a copper-based solid solution, we obtained almost pure iron instead of an iron-based solid solution. A superimposed solution can be stable for a long time due to the same reasons; for its decay, the formation of a solid solution of copper in iron is necessary.

Summarizing the results of the present work and our earlier studies [26,27,28], we concluded that there was an absence of advanced grain boundary diffusion of iron in copper in undiluted copper-based solutions at a wide temperature range (above 550 °C). This could be explained by the negative adsorption of Fe in Cu, which decreased the *p*-value and provoked an additional braking force.

## 5. Conclusions

The grain boundary diffusion of two elements with negative segregation (Ni and Fe) in Cu was studied. The results of the determination of the bulk diffusion coefficients for Ni and Fe demonstrated the sufficient accuracy of the methods used for the GB diffusion studies. The weak concentration dependence of the bulk diffusion coefficient allowed us to use, in this case, the Wipple solution for the GB diffusion problem, with modifications to the initial and boundary conditions. In contradiction to the literary data, it was shown that Ni GB diffusion at a high concentration was significantly slower than in a dilute solution and could be characterized by a high value of effective activation energy of 160 kJ/mol, which could be explained by the negative adsorption of nickel on the Cu GB. There was also no concentration dependence for the iron diffusion, although the studies that were carried out corresponded with supersaturated solid solutions with a maximum concentration 10–15 times higher than the solubility at a given temperature. Regarding the grain boundary diffusion of iron, a significant negative adsorption reduced the depth of the penetration of iron along the GB and explained why there was no advanced grain boundary diffusion.

## Figures and Tables

**Figure 1 materials-16-00922-f001:**
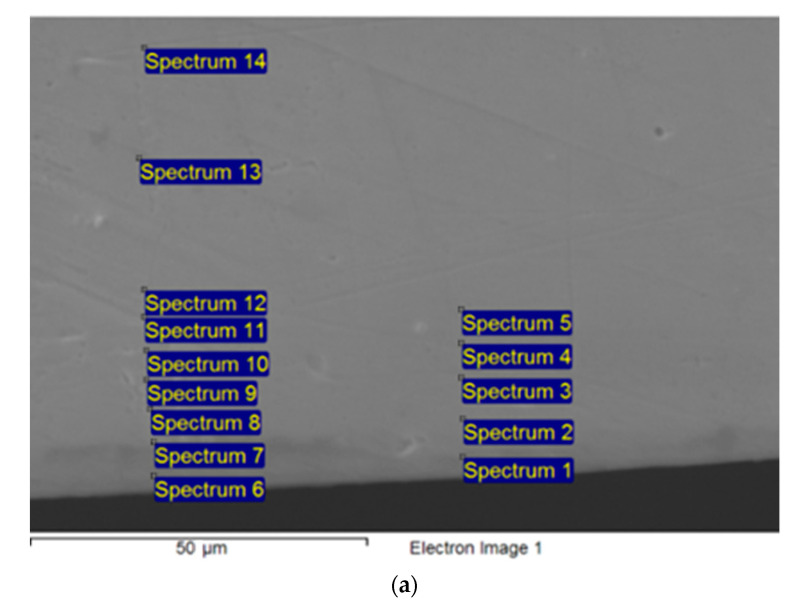
SEM images: (**a**) surface of the cross-section of Cu-Ni diffusion couple; (**b**) the foil after annealing (opposite to Fe layer side).

**Figure 2 materials-16-00922-f002:**
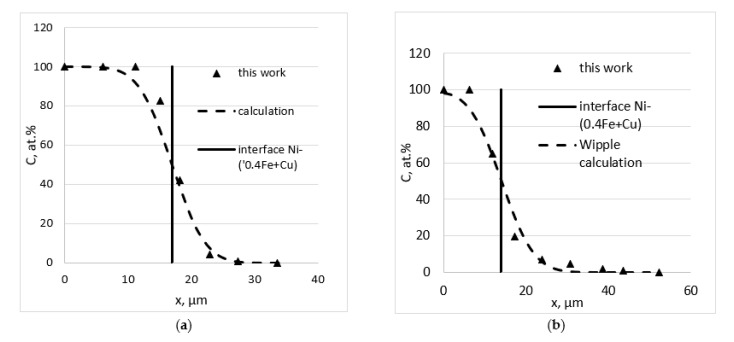
Concentration profiles for bulk (**a**) and GB (**b**) diffusion of Ni in Cu + 0.4 Fe at% at T = 700 °C for 72 h.

**Figure 3 materials-16-00922-f003:**
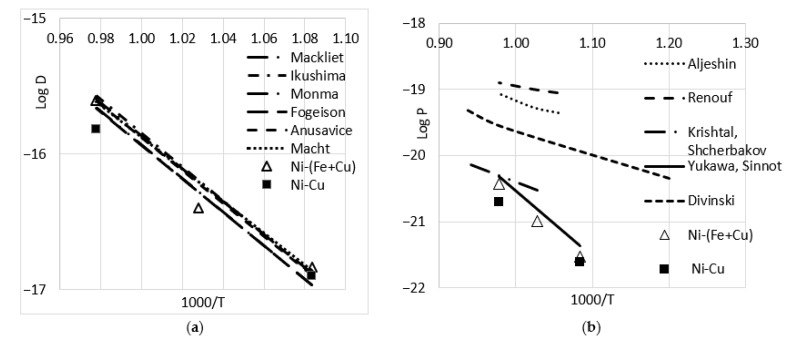
Arrhenius plots for bulk [9,10,11,12,13,14] (**a**) and GB [19,20,21,22] (**b**) diffusion of Ni in Cu, according to different authors. Points correspond with the data obtained in the present work. Averaged value for Cu-Fe alloys are presented as triangles.

**Figure 4 materials-16-00922-f004:**
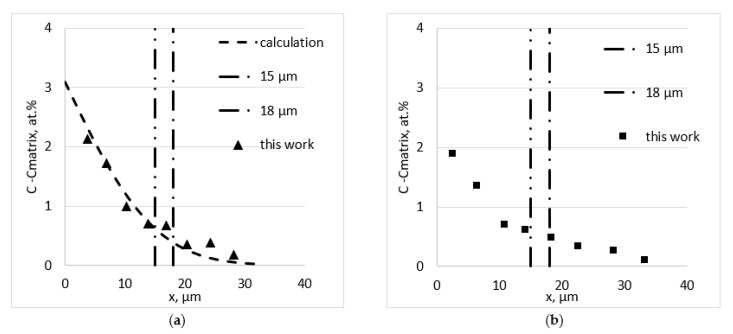
Typical concentration profiles corresponding with measurements far away (**a**) and near to the GB (**b**) for Fe diffusion of Cu + 0.3 Fe at.% (T = 700 °C; t = 49 h).

**Figure 5 materials-16-00922-f005:**
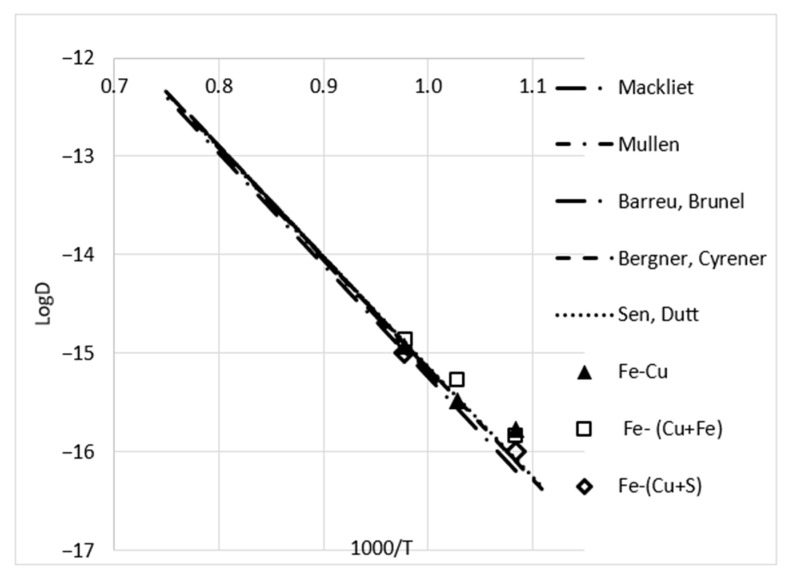
Arrhenius plot for bulk diffusion of Fe in Cu according to different authors [9,15,16,17,18]. Points correspond with the data obtained in the present work.

**Table 1 materials-16-00922-t001:** Ni bulk diffusion coefficient (*D*) and GB diffusion triple product (*P* = *sδD_b_*).

T, °C	Bulk Diffusion Coefficient *D* × 10^16^, m^2^/s			GB Diffusion Triple Product *P* × 10^21^, m^3^/s		
	Pure Cu	Cu + 0.3% Fe	Cu + 0.4% Fe	Pure Cu	Cu + 0.3% Fe	Cu + 0.4% Fe
750	1.5	2.5	2.4	2	5	2.5
700	-	0.41	0.4	-	0.84	1.2
650	0.13	0.18	0.12	0.25	0.23	0.36

**Table 2 materials-16-00922-t002:** Results of determination of *Cs* obtained by the extrapolation of Equation (4) on *x* = *a* at T = 700 °C.

Time	24 h	48 h	72 h	96 h	192 h	312 h
*Cs*, at.%	2.9	3.1	3.2	3.1	2.7	3.4

**Table 3 materials-16-00922-t003:** Bulk diffusion coefficient for Fe-Cu, Fe-(Cu + Fe) and Fe-(Cu + S).

T, °C	Bulk Diffusion Coefficient *D* × 10^16^, m^2^/s					Max. Conc., *Cs*	Solubility, C_0_ [32]
	Pure Cu	Cu + 0.3% Fe	Cu + 0.4% Fe	Cu + 0.002% S [27]	[9]		
750	12	11	18	10.2	12.7	3.1	0.21
700	4.8	5.5	5.4	-	3.4	3.0	0.3
650	1.7	1.2	1.7	1	0.82	2.2	0.46

**Table 4 materials-16-00922-t004:** Calculated and measured concentration (at.%) far from the GB and the measured value near to the GB opposite to the diffusant layer side of the foil.

Fe-Cu	Grain Boundary, at.%	Average Bulk, at.%	Calculation Bulk, at.%
650 °C, 65 h	0.4	0.3	0.3
650 °C, 120 h	0.4	0.3	0.4
700 °C, 24 h	0.3	0.3	0.3
700 °C, 45 h	0.4	0.4	0.5
750 °C, 15 h	0.4	0.4	0.3
750 °C, 27 h	0.4	0.3	0.4
750 °C, 70 h	0.5	0.4	1.0

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
