# Peer review of "Peculiarities of Fe and Ni Diffusion in Polycrystalline Cu"

_materials, 2023, doi:10.3390/ma16030922_

Round 1

Reviewer 1 Report

In this work, the authors determined the experimental diffusion coefficients of Ni and Fe in polycrystalline copper. The paper is written in good English. However, the novelty of the work is low since the same subject has been studied several times before. Furthermore, only an experimental approach to diffusion has been utilized. Modelling is lacking. The paper is highly specialized and focuses on a well-established topic. Therefore, I do not recommend publishing this work in Materials. It should be directed to journals such as Journal of Phase Equilibria and Diffusion or Defect and Diffusion Forum. The authors should consider the following comments:

1.The diffusion of Ni and Fe in Cu has been studied by several authors before (Figs. 1 and 2). The diffusion coefficients measured by the present authors are in agreement with previous studies. Therefore, I don’t see a clear novelty aspect of the present work. If there is a novelty, it must be clearly highlighted.

2.Most references date back to the 1950s - 1970s. The topic of Fe and Ni diffusion in Cu is well-established and does not attract significant attention anymore. Furthermore, the authors themselves studied the Fe diffusion in Cu several times before; see references 23, 24. I do not see a need for the present paper. It only focuses on “pecularities” of Fe and Ni diffusion. If the paper has a concrete value that can be appreciated by a broad audience of Materials, it must be clearly specified.

3.The paper is highly specialized. It should be directed to more specialized journals on diffusion, where it can find its audience.

4.In the Methods section, the authors write that they used the EDX method to determine the concentration profiles (line 81). This method has a low resolution. The data points are relatively sparse, see Figs. 4, 5. The resolution is low even compared to reference [3] from 1958. An EPMA method (microprobe) should have been utilized instead.

5.To determine the grain boundary diffusion coefficient, one must know the grain boundary width (δ). There is no estimation of δ in the present work. Only SEM images with low resolution are provided. The grain boundaries should have been inspected by HR TEM.

6.There is no modelling of Fe and Ni diffusion in Cu and Cu alloys. It is highly recommended to include some simulations of atomic diffusion in the crystal lattice of Cu.

7.Experimental conditions, i.e., temperatures and annealing times, should be specified in the abstract. Furthermore, the results obtained, that is, diffusion coefficients, should be listed.

8.Figs. 1 and 2, i.e., the confrontation of the present results with previous papers, should be moved to discussion.

9.The importance of Fe and Ni diffusion in Cu and Cu alloys must be discussed in the paper. Any technological applications of the materials at high temperatures should be highlighted.

Author Response

Dear reviewer!

Thank You very much for reading the text and your comments. We tried to change the text taking into account your opinion. We believe that some points  must be clarified additionally: 

# 1, 2 and 9: Bulk diffusion coefficient determination was necessary only in order to check the accuracy of the method and possibility to use the developed mathematical treatment.

General interest to chosen system is connected with negative segregation which can be seen in these systems. At other hand the application is quite obvious. It is connected with necissity to describe the phase growth separately in the grain bulk and at GB. Some specific application was added to the text.

#4. As for method we have writen EDS, but it is EPMA with energy dispersed analysis. We corrected the name. The step for concentration measurements was taken 5 microns because typical zone of analysis is around 3 microns, and decreasing the step almost does not change the accuracy.

# 5 The macroscopic GB flux is determined by the triple product P (see e.g. ref. 1 and 2) even in the case of particle growth at GB (ref. 28) it is P value determines the growth. That is why it is no need to define GB width separately. Besides, according e.g. S. Divinsky diffusion GB width can be taken as 0.5 nm for most part of materials, including metals and oxides.

#6: As for modelling (e.g. MD) such analysis was also made already, including our papers. In fact it is quite difficult to compare the directly the results of simulation with experiments. Here we use them only as for discussion.

8 It seems better to present the literature data in first section in order to demonstrate the problem: high scatter of the results of GB diffusion while we have very good agreement for bulk diffusion. And to formulate the problem: in dependence of the task you need to solve, you must take the different values. Unfortunately, up to now, we have no other way to describe the process.

Reviewer 2 Report

Paper deals with the study of diffusion in bulk and grain boundaries in Fe-Cu and Ni-Cu systems.

It brings new experimental data, so it deserves to be published.

Before publication some formal improvements are recommended:

  1. Line 73: It should be: “... [21] more than 1... “

  2. Line 133: Explain, please, the difference with data (which data create the difference)

  3. Line 147: It should be: “ ...far away... “

  4. Line 188: Explain, please, what should not be expected

  5. Line 202: It should be: “ Calculated value (160 kJ/mol) is significantly higher than…”

  6. Line 233: It should be: “ ...solution. Superimposed… “

Author Response

Tnak you very much for attentive reading.

Reviewer 3 Report

The comments can be drawn as follows:

1.Repetitive use of similar sentences in introduction and abstract. Objectives of the work is not mentioned clearly. Grammatical errors and use of hyphenate is observed in the article.

2.Recent literature has not been highlighted

A. "Effect of cutting parameters on tool wear, cutting force and surface roughness in machining of MDN431 alloy using Al and Fe coated tools" Materials Research Express 6 (1), 016401

B. "Tribological behaviour of monolayer and multilayer Ti-based thin solid films deposited on alloy steel"Materials Research Express 6 (2), 026419

3. Materials and methods explanation is not clear.

4. Explanation of the results and discussion needs to be improvised, no experimental standards are used (ASTM or ASM)

Author Response

Thank you very much for attentive reading and your suggestion. We add some ref. for possible application and rewrite some parts of description.

Round 2

Reviewer 1 Report

The authors partially answered my previous comments in the cover letter. However, the paper itself still requires a major revision.

1.You haven’t highlighted the changes in the manuscript itself. Therefore, it was difficult to judge whether you really followed the previous comments or not. It is not enough to respond to the reviewer’s comments in the cover letter alone. These must be reflected in the paper itself. All changes must be clearly labelled with either a yellow background or a different color in the manuscript itself.

2.The manuscript must include a clear novelty statement. The novelty statement must be placed at the end of the introduction.

3.It is very unusual to present the actual results in the introduction (Fig. 1). It leaves the reader wondering what the purpose of the present article is. It seems that you have published the diffusion coefficients of Fe in Cu previously (ref. [26], [27]). Therefore, the present paper can be regarded as redundant. You need to clearly substantiate the reason for the current paper. If you want to compare the behavior of Fe with Ni, state it clearly. You can also change the title of the manuscript to make Ni stand out. The paper must have a clear novelty aspect, either a new material studied and/or new technique used. The paper must present new results. Otherwise, it cannot be accepted for publication.

4.Figures are sloppily prepared:

Fig. 3a contains a Russian script. The Russian script must be removed.

The titles of the axes in several plots (Fig. 5 a, b) are overlaid with numbers. Please, take more care when preparing the figures. All axes must be readable.

Figure 1b contains two solid lines (Renouf data and present data). The lines must be differentiated. Use different colors to differentiate them.

5.All tables use decimal commas. The commas must be replaced with decimal points.

Author Response

  1. I agree and apolagize for that in this version I indicates by yellow the main changes. Some small changes (e.g. reference numeration, etc.) are not indicated directly.
  2. it is re- writen
  3. Here we enlarge the time - temperature range of the diffusion study and summarize two approach. We use the aproach with foils in order to avoid some effect of sample preparation on measured mass-transport tyo be sure in main effect. In last verion the figure 1 and 2 were removed in results section so the structure of article was changed. As for Ni behaviour- firstly we do thought about Ni as just demonstration of correctnes of measurements, but further we studied - the more we saw the strange results. That is the reason why we include these data. 
  4. The pictures were modified.
  5. it is corrected.